# Long-Term Outcomes after Transcatheter Mitral Valve-in-Valve or Valve-in-Ring Procedures

**DOI:** 10.3390/jpm13050803

**Published:** 2023-05-08

**Authors:** Manuel Wilbring, Asen Petrov, Sebastian Arzt, Julia Patricia Eiselt, Ali Taghizadeh-Waghefi, Klaus Matschke, Utz Kappert, Konstantin Alexiou

**Affiliations:** Department of Cardiac Surgery, University Heart Center Dresden, Fetscherstrasse 76, 01307 Dresden, Germany

**Keywords:** transcatheter mitral, valve-in-valve, valve-in-ring

## Abstract

Background: Redo mitral valve surgery is the standard of care for failed mitral bioprostheses or recurrence of mitral regurgitation after repair. Nonetheless, catheter-based valve-in-valve (ViV) or valve-in-ring (ViR) procedures have increasingly become viable alternatives in high-risk subpopulations. Despite reported good initial results, little is known about longer-term outcomes. Here, we report the long-term outcomes of transcatheter mitral ViV and ViR procedures. Methods: All consecutive patients (*n* = 54) undergoing transcatheter mitral ViV or ViR procedures for failed bioprostheses or recurring regurgitation after mitral repair in the time period between 2011 and 2021 were retrospectively enrolled. The mean age was 76.5 ± 6.5 years, and 30 (55.6%) of the patients were male. The procedures were done using a commercially available balloon-expandable transcatheter heart valve. Clinical and echocardiographic follow-up data were obtained from the hospital’s database and analyzed. Follow-up reached up to 9.9 years with a total of 164.3 patient-years. Results: A total 25 patients received a ViV and 29 patients a ViR procedure. Both groups were at high surgical risk with an STS-PROM of 5.9 ± 3.7% in ViV and 8.7 ± 9.0% in ViR patients (*p* < 0.01). The procedures themselves were mainly uneventful with no intraoperative deaths and a low conversion rate (*n* = 2/54; 3.7%). VARC-2 procedural success was low (ViV 20.0% and ViR 10.3%; *p* = 0.45), which was either driven by high rates of transvalvular pressure gradients “>5 mmHg” (ViV 92.0% and ViR 27.6%; *p* < 0.01) or residual regurgitation “>trace” (ViV 28.0% and ViR 82.7%; *p* < 0.01). ICU-stay was prolonged in both groups (ViV 3.8 ± 6.8 days and ViR 4.3 ± 6.3 days; *p* = 0.96) with acceptable hospital stay (ViV 9.9 ± 5.9 days and ViR 13.5 ± 8.0 days; *p* = 0.13). Despite 30-day mortality being acceptable (ViV 4.0% and ViR 6.9%; *p* = 1.00), the mean posthospital survival time was disappointingly low (ViV 3.9 ± 2.6 years and ViR 2.3 ± 2.7 years; *p* < 0.01). Overall survival in the entire group was 33.3%. Cardiac reasons for death were frequent in both groups (ViV 38.5% and ViR 52.2%). Cox-regression analysis identified ViR procedures as a predictor of mortality (HR 2.36, CI 1.19–4.67, *p* = 0.01). Conclusions: Despite acceptable immediate outcomes in this high-risk subpopulation, long-term results are discouraging. Transvalvular pressure gradients as well as residual regurgitations remained drawbacks in this real-world population. The indication for catheter-based mitral ViV or ViR procedures rather than conventional redo-surgery or conservative treatment must be thoughtfully considered.

## 1. Introduction

With the rise of catheter-based procedures, the treatment of valvular heart disease has experienced profound changes during the past twenty years [1]. In 2002 Alain Cribier reported the first successful transcatheter aortic valve implantation in a human and in 2008, Walther, Kempfert, and colleagues described the principle of transcatheter valve-in-valve therapies [2,3,4]. In 2009 Cheung and colleagues adopted the valve-in-valve concept to the mitral valve and in 2012 and 2013 the Dresden group added early experiences with mitral valve-in-valve and valve-in-ring procedures [5,6,7]. Despite the fact that re-operative mitral valve surgery has been proven to be—at least in experienced hands—as safe as primary mitral surgery, catheter-based alternatives have some promising aspects, especially in high-risk or inoperable subpopulations [5,6,8]. The early experiences confirmed the short procedure times of a straightforward and less invasive catheter-based treatment strategy; however, nothing is known about durability or long-term outcomes [5,6]. Basically, this has not changed during the past decade. Despite the international and multicentric valve-in-valve registry (VIVID) (which was set up by Danny Dvir) confirming favorable immediate outcomes, there is still little known beyond one year of follow-up [9,10,11,12].

This gap in knowledge needs to be closed. Mitral valve disease is one of the most common valvular heart diseases and it is estimated that mitral regurgitation is the second most prevalent disease after aortic stenosis in adult patients with severe VHD [13]. Of course, the current gold standard of treatment is surgical repair or replacement [1]. However, as life expectancy increases, we can expect a rise in the need for mitral valve reinterventions in failed repairs or degenerated prostheses.

For an adequate and evidence-based decision-making process, it is necessary that more data must be gathered—especially beyond the well-described immediate and 1-year success. The present study sought to close the mid- to long-term data gap by adding follow-up of patients undergoing transcatheter mitral valve-in-valve or valve-in-ring procedures.

## 2. Patients and Methods

### 2.1. Study Design and Inclusion and Exclusion Criteria

The study was designed as a retrospective data analysis. The study population consisted of all consecutive patients who underwent a transcatheter mitral valve-in-valve or valve-in-ring procedure at our facility between September 2011 and December 2021 (*n* = 54). During the same time period, a total of 4718 patients were treated by transcatheter heart-valve procedures. Mitral valve-in-valve or valve-in-ring procedures accounted for 1.1%.

Accordingly, main inclusion criteria were presence of a failed biological mitral valve or recurrent mitral regurgitation after mitral repair using an annuloplasty ring as well as a Heart Team decision for a catheter-based procedure. This decision was usually based on high surgical risk according to comorbidities or frailty, in accordance with the past and present guidelines [1].

Exclusion criteria were active endocarditis or emergent or salvage surgery. Concomitant procedures other than the mitral valve surgery at the index procedure as well as a history of endocarditis were not considered as exclusion criteria. 

### 2.2. Data Collection, Ethic Statement, and Study Endpoints

All patient data were anonymized before accessing the database. The institutional ethics committee approved the inclusion of the patients and waiver for the requirement for informed consent was granted (EK 53022010). 

Study endpoints were specified in accordance with the Mitral Valve Academic Research Consortium criteria [14]. Risk scores (EuroSCORE II and STS-PROM score) were calculated for a redo surgical mitral valve procedure [15,16]. Follow-up data were obtained from all recorded patient contacts and available echocardiographic examinations.

### 2.3. Patients, Study Groups, and Follow-Up

A total of 54 patients according to the inclusion and exclusion criteria were included in the present study. The mean age was 76.5 ± 6.5 years, and 30 (55.6%) of the patients were male. Mean STS-PROM was 7.42 ± 4.24; the EuroSCORE II averaged 14.4 ± 8.72. The overall follow-up reached up to 9.9 years with a mean follow-up time of 4.8 ± 0.6 years. A total of 164.3 patient-years was analyzed.

According to the type of procedure, the study group was divided into two subpopulations: (a) valve-in-valve (ViV) procedures including *n* = 25 patients and (b) valve-in-ring (ViR) procedures including *n* = 29 patients.

The number of cases per year was 2.7 ± 1.4 overall, 2.3 ± 0.9 in the ViV and 3.2 ± 1.7 in the ViR group. A special assessment of potential learning-curve effects was not performed due to the limited annual caseload.

### 2.4. Statistical Analysis

Descriptive and analytic statistical analysis was performed. Continuous variables were expressed as means and standard deviations. Categorical variables were summarized as counts and percentages. A *p*-value under 0.05 was considered significant. Student’s *t*-test, Chi-squared test, and Fisher’s exact test was used to compare demographic and postoperative data between the valve-in-valve (ViV) and the valve-in-ring (ViR) group. The Shapiro–Wilk test was used to assess the normality of the continues variables, and when abnormal distributions were detected, the Mann–Whitney non-parametric test was performed. Kaplan–Meier curves were used to demonstrate postoperative survival and time between the initial procedure and the reintervention. Univariable and multivariable Cox regression was conducted to evaluate possible risk factors for postoperative mortality. All the analyses were performed using R software, version 4.2.1 (R Foundation for Statistical Computing) [17]. Results are presented in written form and summarized in tabular and graphic form.

## 3. Results

### 3.1. Baseline Characteristics

Due to the differences in indications, both groups basically showed good comparability concerning their baseline characteristics risk profile. Nonetheless, some risk-related items were not counterbalanced between ViR and ViV patients.

Age and sex did not differ significantly between the ViV (77.4 ± 6.3 years; 52.0% male) and the ViR patients (75.8 ± 6.7 years; 58.6% male) (*p* = 0.35 and *p* = 0.83, respectively). Cardiovascular risk factors such as arterial hypertension, diabetes mellitus, dyslipidemia, and further pre-existing conditions in the medical history were equally distributed in both groups. Patients undergoing a ViV procedure had a significantly lower rate of preoperative history of permanent pacemaker implantation of 20.0% (*n* = 5) compared to 69.0% (*n* = 20) in the ViR group (*p* < 0.001).

The calculated risk scores were consistently higher in the ViR group with a EuroSCORE II averaging 16.9 ± 9.0%, and a mean STS-PROM score of 8.7 ± 4.3% compared to 11.5 ± 7.5% and 5.9 ± 3.7% in ViV patients (both *p* = 0.01). There was no significant difference in the time-interval between the index procedure and the present intervention. In ViV patients, the mean was 9.3 ± 3.8 years and in ViR patients it was 7.7 ± 5.0 years (*p* = 0.20) (Figure 1).

The indications for the index cardiac procedure differed significantly between both groups. In the ViV cohort the index procedure was more frequently performed due to structural mitral disease (84.0%, *n* = 21; *p* < 0.01), whereas functional mitral disease was the most frequent indication for surgery amongst ViR patients (72.4%, *n* = 21; *p* < 0.01). 

Meanwhile, the overall time to reintervention for the entire group is depicted in Figure 1. The time to reintervention did not differ significantly when comparing the indication for the initial procedure (functional compared to structural mitral valve disease; *p* = 0.98). Table 1 summarizes patient baseline characteristics.

### 3.2. Echocardiographic Baselines

The ViR group showed a significantly worse left ventricular ejection fraction of 39.3 ± 15.1% compared to 53.6 ± 12.4% in the ViV cohort (*p* < 0.001). This finding was consistent with a higher LVEDD in ViR patients (49.1 ± 6.5 mm vs. 58.6 ± 7.8 mm, *p* < 0.001). Right ventricular function and distribution of the severity of mitral regurgitation were similar in both groups (Table 2). There was a higher average peak (27.4 ± 6.4 mmHg vs. 18.4 ± 6.6 mmHg; *p* < 0.01) and mean gradients (10.7 ± 3.5 mmHg vs. 6.3 ± 2.7 mmHg; *p* < 0.001) observed in ViV patients compared to the ViR cohort. This correlated with the higher incidence of moderate or severe stenosis (64.0% vs. 20.6%) in the ViV cohort as the primary type of failure. Preoperative echocardiographic data are summarized in Table 2.

### 3.3. Procedural Data

The stented tissue valve implanted during the index procedure of the ViV patients had an average labeled size of ViV 29.7 ± 1.1 mm. In the ViR cohort, the majority of annuloplasty rings implanted during the index procedure were semirigid (*n* = 28; 96.6%). In one single case (*n* = 1; 3.4%), an incomplete flexible ring had been used. The mean labeled size of the priorly implanted annuloplasty rings was ViR 28.4 ± 1.5 mm. During the actual procedure, all patients received a balloon-expandable transcatheter prosthesis. Mean labeled size was 27.9 ± 1.5 mm in the ViV and 25.9 ± 1.7 mm in the ViR cohort, which differed significantly (*p* < 0.001). There was no significant difference between the groups in terms of access, prosthesis model, surgery time, rate of conversion, intraoperative complications, or procedure success. According the M-VARC criteria, the procedural success was low in both groups, which was mainly driven by increased transvalvular pressure gradients (Table 3). When postprocedural mitral regurgitation was present, it predominantly consisted of paravalvular, and, less frequently, transvalvular regurgitation. Left ventricular outflow tract obstruction was rare and occurred in 4.0% of the ViV and 6.9% of the ViR patients. Table 3 summarizes procedural data.

### 3.4. Postoperative Course and Hospital Outcomes

Postoperative course and hospital outcomes were mainly comparable between both groups—the primary ICU-stay did not differ significantly with 3.8 ± 6.8 days in the ViV cohort and 4.3 ± 6.3 days in the ViR cohort (*p* = 0.96). The same was true for hospital stay (ViV: 9.9 ± 5.9 days vs. ViR 13.5 ± 8.0 days, *p* = 0.13) and 30-day mortality with 4.0% (*n* = 1) in the ViV and 6.9% (*n* = 2) in the ViR group (*p* = 1.00). The postoperative data are summarized in Table 4.

### 3.5. Follow-Up Data

Survival analysis of the entire group showed a median overall survival of 3.9 years. The respective survival rates were 70.1 ± 6.3% at one year, 32.7 ± 7.0% at five years, and 30.0 ± 6.9% at seven years (Figure 2).

When comparing survival by type of procedure (ViV vs. ViR), median survival was longer in the ViV group with 4.4 versus 1.2 years, respectively (*p* = 0.01). Higher survival rates in ViV patients were also distinct at one year (91.8% vs. 51.7%), five years (42.9% vs. 23.3%), and seven years (35.7% vs. 23.3%), as depicted in Figure 3 (*p* = 0.012).

This coherence was independent from the risk stratification. If stratified after STS-PROM with separate consideration of a lower risk-strata (STS-Score < 8%) and high risk-strata (STS-Score ≥ 8%), the observed significantly higher mortality in the ViR group persisted (Figure 4).

A significant reduction in mitral regurgitation in the postprocedural echocardiographic control was recorded. Echocardiographic follow-up after one year demonstrated that the initial results concerning regurgitation or stenosis were sustained independently of the performed type of procedure (Figure 5).

We did not observe significant changes in LVEF in follow-up measurements. Significant improvements in NYHA class at 1-year follow-up were observed in both groups, but NYHA functional class appeared to deteriorate again after 5 years of follow-up (Figure 6).

### 3.6. Survival Analysis

Univariate Cox regression was used for identifying possible predictors of long-term survival. Herein, valve-in-ring as type of procedure, was found to be a significant predictor of mortality with a hazard ratio (HR) of 2.36 (95% CI 1.19–4.67; *p* = 0.01) together with female sex (HR 2.14 95% CI 1.07–4.27; *p* = 0.03), surgery duration (HR 1.02; 95%CI 1.01–1.03; *p* < 0.001), and conversion to sternotomy (HR 29.33; 95% CI 4.82–178.4; *p* < 0.001). The transfemoral access routes showed a trend of inferior outcomes (*p* = 0.06) in univariate Kaplan–Meier analysis (Figure 7), which could not be confirmed in multivariate assessment (*p* = 0.43). Table 5 summarizes univariate risk factors for all-cause mortality during follow-up.

Multivariable regression analysis only identified valve-in-ring as type of procedure (HR 2.36 (95% CI 1.19–4.67; *p* = 0.01), female sex (HR 2.49; 95%CI 1.19–5.20; *p* = 0.02), and conversion to sternotomy (HR 22.49; 95% CI 3.47–145.92; *p* = 0.001) as mortality factors. Additionally, EuroSCORE II was conformed to be predictive for mortality. Table 6 summarizes the results of the multivariate Cox-regression analysis.

## 4. Discussion

Concerning the immediate outcomes, transcatheter mitral valve implantation is a viable alternative in high-risk patients presenting with a degenerated bioprosthesis or recurrent regurgitation after mitral repair. The observed 30-day mortality proved to be lower, or at least comparable, to the predicted mortality by scores [11,12,18]. These findings could be confirmed by our current study. Here, the observed 30-day mortality was also lower than predicted mortality, which was true for the type of procedure (ViV or ViR) as well as for both used scoring systems (STS-PROM and EuroSCORE II). Nonetheless, the 30-day mortality remained substantial. This has to be interpreted with regard to a real high-risk population, consisting of patients rejected for a redo surgical procedure. With that background, the comparatively low 30-day mortality rates are acceptable.

Despite certain risk factors being more pronounced in ViR patients (higher EuroSCORE II and STS-PROM score, and a lower ejection fraction), we did not identify any of these factors to be an independent predictor of inferior outcome. Intraprocedural and postprocedural outcomes did not differ significantly between groups. Similarly to previously reported low success rates, device success was low in both groups [12,18]. As reported by Simonato et al., the low device success and low subsequent procedure success rates were mainly driven by low mean transvalvular gradients, being mostly ≥5 mmHg in postinterventional measurements. When modified criteria were used (mean gradient ≤10 mmHg), the procedural success rates were notably higher, but still not comparable to what is common after a surgical redo-procedure.

However, as previously observed in similar studies, these initially encouraging results worsen in the long-term. This was particularly true for patients undergoing a ViR procedure. Our observations of a significantly shorter median survival in patients who had previously undergone a mitral annuloplasty confirmed these findings. Hu et al., in a systematic review of literature, reported six-month mortality rates of 18.5% for ViV patients and 38.5% for ViR patients, despite the fact that mean overall in-hospital mortality was satisfactory at 5.8% [19]. The inferior outcome of ViR persisted when subgroups were evaluated separately—patients with reduced ejection fraction, patient with preserved ejection fraction, patients with high-risk scores, and patients with low-risk scores. With regard to all inherent limitations of this study, the results might indicate that, independent of other risk factors, the implantation of a transcatheter aortic valve prosthesis in a mitral annuloplasty ring is associated with higher long-term mortality of any cause compared to the implantation of the same prosthesis in a previously implanted biological valve prosthesis. This finding confirms the results from other studies [12,19].

The key aspect distinguishing our study from previous ones is that we present a single-center high-volume experience with mid- to long-term follow-up. By reporting on all consecutive patients who underwent this transcatheter procedure we can avoid the selection bias that can occur in registries [12,18]. Furthermore, despite the overall lower number of cases in the present series compared to multicenter studies, we have a much higher single-center caseload. For example, Guerrero et al. reported low center caseloads as 4.22 ViV and 2.12 ViR cases per hospital in 172 hospitals. The low center caseload in registry patients could mitigate to some extent the inherent effects of a learning curve [12,18].

It has been previously suggested that the inferior long-term survival in ViR patients may be attributable to one or multiple factors—preoperative comorbidities and cardiac dysfunction, higher risk of potential LVOT obstruction, or mismatch in the shape of the annuloplasty ring and the transcatheter aortic valve prosthesis. Cardiac dysfunction may be more pronounced in the annuloplasty group, due to the hesitancy to operate on a previously reconstructed mitral valve. However, there is no way to measure the interval between the time at which an intervention would have been indicated and the time it was performed and compare it in both groups. Therefore, a longer delay in surgery and subsequently more severe deterioration in cardiac function in the ViR group can neither be confirmed nor ruled out. An LVOT obstruction is theoretically more likely in ViR patients as the anterior mitral leaflet is still present in all patients and comes in direct contact with the newly implanted prosthesis, whereas in ViV patients it has already been either resected or is being retracted by the initial valve prosthesis. Our data do not support this thesis because of the overall low rate of LVOT obstruction.

It remains that the inability of the circular valve to adapt to the irregular D-shaped form of the annuloplasty ring probably causes hemodynamic and functional deficits is less effective in removing the predominant cause of failure, which in the long-term leads to a diminished prognosis [18,20]. This assumption is supported by our finding, that postprocedural mitral regurgitation as well as paravalvular mitral regurgitation were significantly more frequent in the ViR group. However, whether this is causative for the long-term outcomes remains speculative and warrants further investigation.

## 5. Limitations

The present study had some inherent limitations. It was a retrospective observational study with a relatively small sample size. This resulted in a relative lack of statistical power, especially because the number patients at risk for whom follow-up time extended beyond four years was quite low. The nature of these procedures, as an individual treatment-attempt, caused impaired comparability, especially in a worldwide context. Finally, the two treatment groups had limited comparability with respect to their baselines.

## 6. Conclusions

ViV and ViR procedures are primarily safe and effective in the short-term. Nonetheless, results of this study, as well reviewed literature, raise concerns about durability in the long-term. In particular, ViR seems to be less effective in removing the primary mode of failure, which in the long-run, potentially causes inferior clinical outcomes.

Nonetheless, the observations made have to be carefully interpreted in the context of an old and high-risk population of treated patients.

The results of the present series also indicate that a note of caution should extend towards younger or even lower-risk patients.

## Figures and Tables

**Figure 1 jpm-13-00803-f001:**
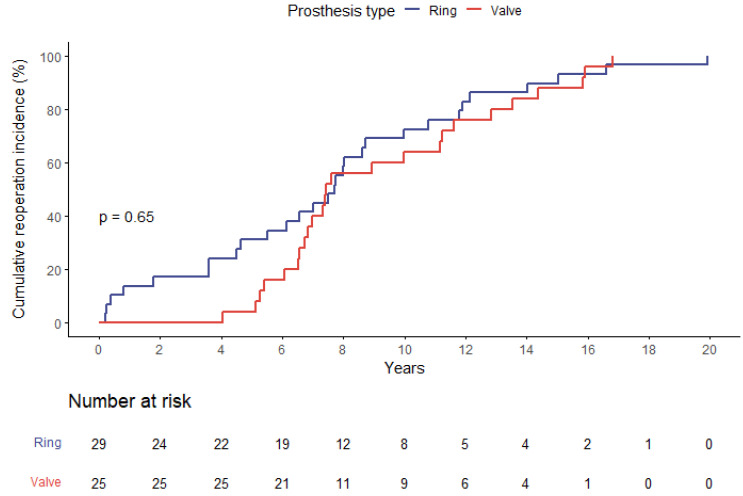
Cumulative incidence of reoperation (by means of ViV or ViR procedure) since the index mitral valve surgery. (Please note: Since this graph exclusively reports patients undergoing reintervention, general failure rates of mitral repair or replacement cannot be deduced).

**Figure 2 jpm-13-00803-f002:**
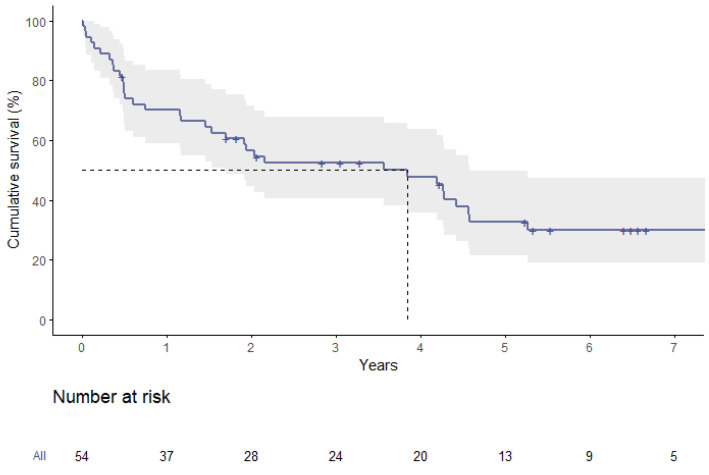
Survival plot of the entire cohort (ViV plus ViR).

**Figure 3 jpm-13-00803-f003:**
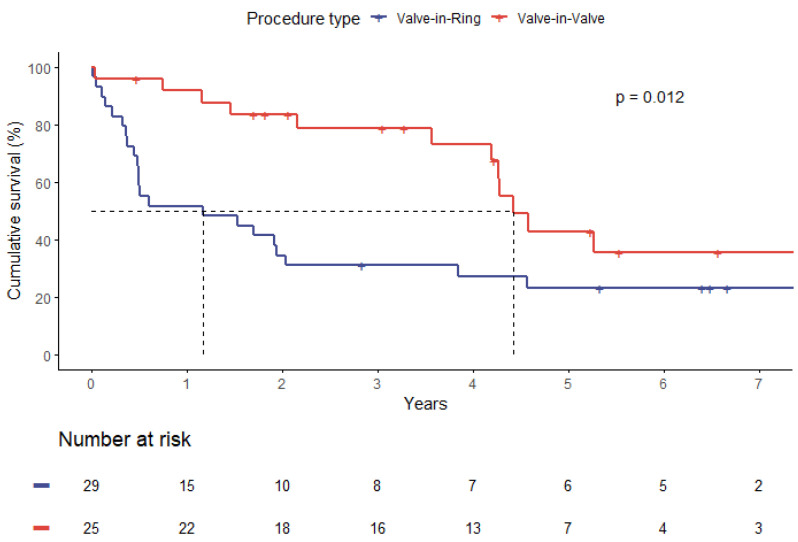
Survival plot by type of procedure (ViV—blue line, ViR—red line).

**Figure 4 jpm-13-00803-f004:**
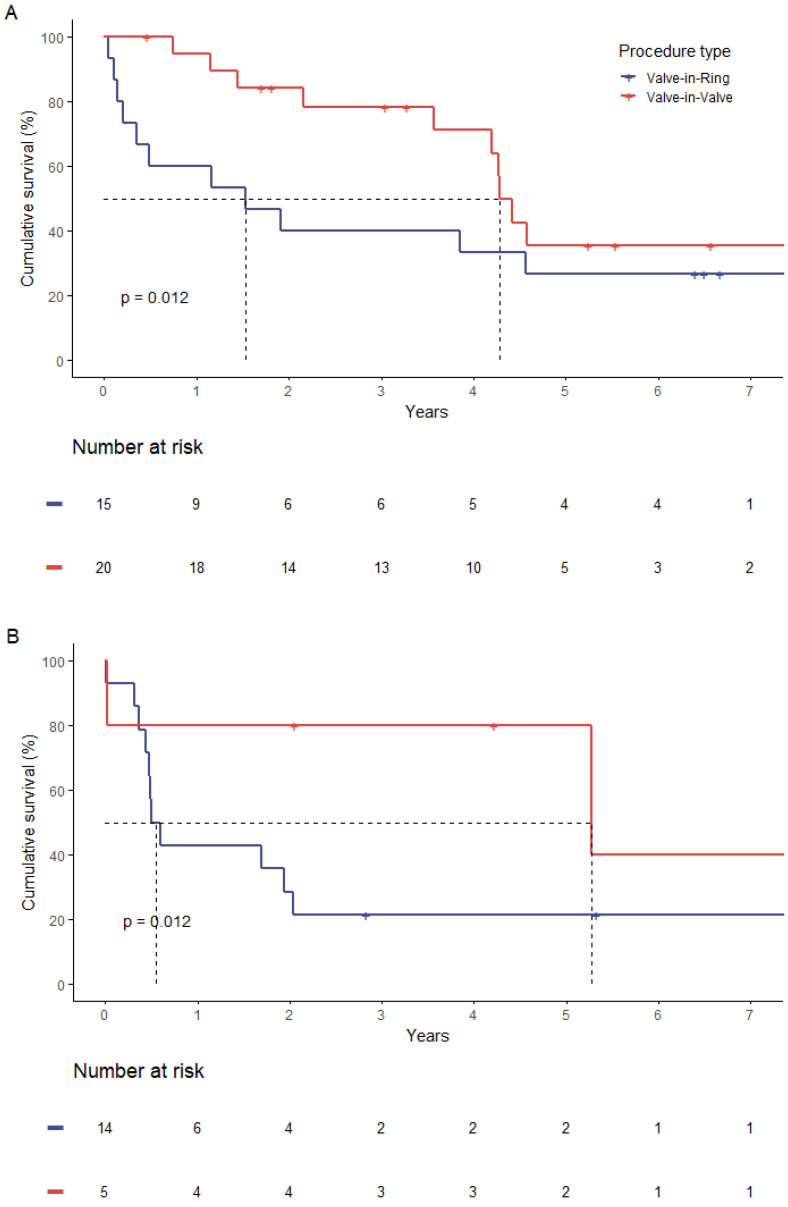
Comparison in survival by type of procedure in patients with low STS-PROM score < 8% (**A**) and high STS-PROM score > 8% (**B**).

**Figure 5 jpm-13-00803-f005:**
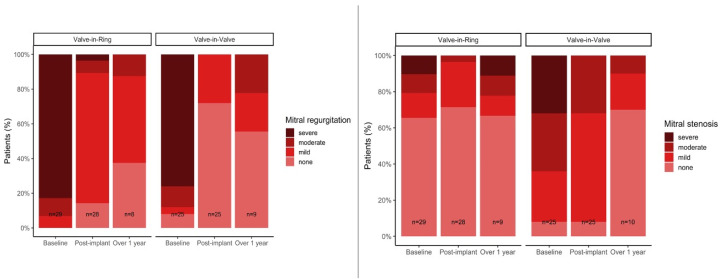
Mitral regurgitation (**left**) and stenosis (**right**) from baseline to follow-up in dependance from the performed procedure.

**Figure 6 jpm-13-00803-f006:**
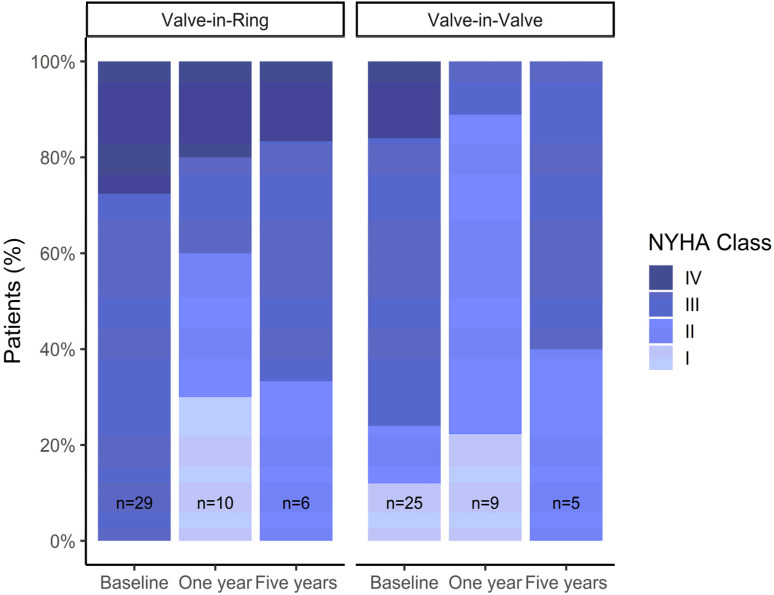
NYHA functional class during follow-up depending from the type of procedure.

**Figure 7 jpm-13-00803-f007:**
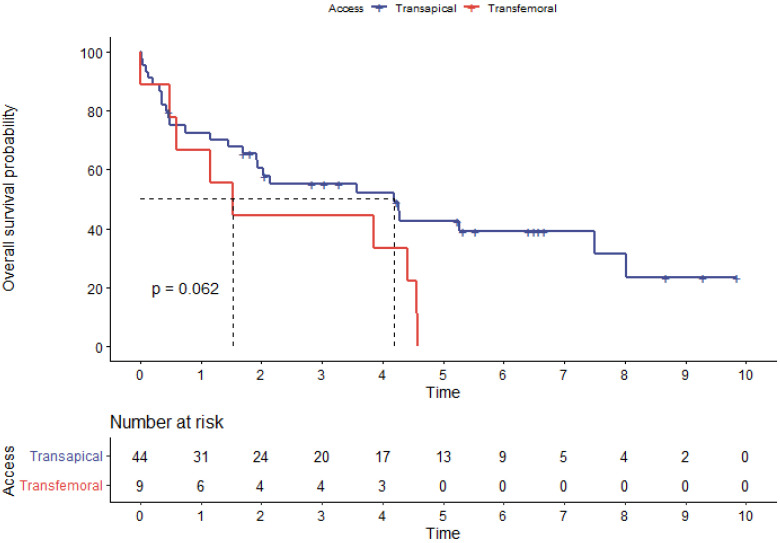
Overall survival probability estimates grouped by access route.

**Table 1 jpm-13-00803-t001:** Demographic data.

	Valve-in-Valve (*n* = 25)	Valve-in-Ring (*n* = 29)	*p*-Value
Age (years)	77.4 ± 6.3	75.8 ± 6.7	0.35
Gender male	13 (52.0%)	17 (58.6%)	0.83
BMI (kg/m^2^)	25.4 ± 4.5	28.1 ± 5.5	0.08
Arterial hypertension	24 (96.0%)	27 (93.1%)	1.00
Diabetes mellitus	9 (36.0%)	11 (37.9%)	1.00
Dyslipidemia	16 (64.0%)	20 (69.0%)	0.92
Coronary artery disease	15 (60.0%)	19 (65.5%)	0.89
Chronic obstructive lung disease	6 (24.0%)	4 (13.8%)	0.49
Pulmonary arterial hypertension			
None	4 (16.0%)	6 (20.7%)	
Moderate	7 (28.0%)	16 (55.2%)	0.06
Severe	14 (56.0%)	7 (24.1%)	
Chronic kidney disease	21 (84.0%)	29 (100%)	0.04
Preoperative dialysis	1 (4.0%)	1 (3.4%)	1.00
GFR (ml/min)	53.0 ± 21.2	44.5 ± 19.0	0.12
Peripheral arterial disease	6 (24.0%)	4 (13.8%)	0.49
History of stroke			0.54
None	21 (84.0%)	27 (93.1%)	
TIA	2 (8.0%)	1 (3.4%)	
Stroke	2 (8.0%)	1 (3.4%)	
Atrial fibrillation	20 (80.0%)	24 (82.8%)	1.00
Preoperative pacemaker	5 (20.0%)	20 (69.0%)	<0.001
NYHA Class			0.04
I	3 (12.0%)	0 (0%)	
II	3 (12.0%)	0 (0%)	
III	15 (60.0%)	21 (72.4%)	
IV	4 (16.0%)	8 (27.6%)	
EuroSCORE II (%)	11.5 ± 7.5	16.9 ± 9.0	<0.01
STS-PROM score (%)	5.9 ± 3.7	8.7 ± 4.3	<0.01
Years since index operation	9.3 ± 3.8	7.7 ± 5.0	0.200
Initial operation for endocarditis	7 (28.0%)	0 (0%)	<0.01
Initial operation indication			<0.001
Structural mitral disease	21 (84.0%)	8 (27.6%)	
Functional mitral disease	4 (16.0%)	21 (72.4%)	
History of coronary bypass	6 (24.0%)	7 (24.1%)	1.000
Size of initial prosthesis (mm)	29.7 ± 1.1	28.4 ± 1.5	<0.001

**Table 2 jpm-13-00803-t002:** Preoperative echocardiographic baseline data.

	Valve-in-Valve (*n* = 25)	Valve-in-Ring (*n* = 29)	*p*-Value
Ejection fraction (%)	53.6 ± 12.4	39.3 ± 15.1	<0.001
LVEDD (mm)	49.1 ± 6.5	58.6 ± 7.8	<0.001
Right ventricular function			0.56
Normal	8 (32.0%)	9 (31.0%)	
Mildly reduced	6 (24.0%)	9 (31.0%)	
Moderately reduced	9 (36.0%)	7 (24.1%)	
Severely reduced	2 (8.0%)	4 (13.8%)	
Mitral regurgitation			0.62
None	2 (8.0%)	0 (0%)	
Mild	1 (4.0%)	2 (6.9%)	
Moderate	3 (12.0%)	3 (10.3%)	
Severe	19 (76.0%)	24 (82.8%)	
Mitral stenosis			<0.001
None	2 (8.0%)	19 (65.5%)	
Mild	7 (28.0%)	4 (13.8%)	
Moderate	8 (32.0%)	3 (10.3%)	
Severe	8 (32.0%)	3 (10.3%)	
Mitral valve area (cm^2^)	1.3 ± 0.6	2.4 ± 1.2	0.16
Peak gradient (mmHg)	27.4 ± 6.4	18.4 ± 6.6	<0.001
Mean gradient (mmHg)	10.7 ± 3.5	6.3 ± 2.7	<0.001

**Table 3 jpm-13-00803-t003:** Operative data.

	Valve-in-Valve (*n* = 25)	Valve-in-Ring (*n* = 29)	*p*-Value
Indication for redo procedure (according to the leading pathology)			0.76
Mitral regurgitation	18 (72.0%)	23 (79.3%)	
Mitral stenosis	7 (28.0%)	6 (20.7%)	
Access			0.48
Transapical	22 (88.0%)	22 (75.9%)	
Transfemoral	3 (12.0%)	6 (20.7%)	
Right anterolateral minithoracotomy (transatrial)	0 (0%)	1 (3.4%)	
Prosthesis model			1
Sapien XT	8 (32.0%)	10 (34.5%)	
Sapien 3	17 (68.0%)	19 (65.5%)	
Prosthesis labeled size (mm)	27.9 ± 1.5	25.9 ± 1.7	<0.001
Surgery time (min)	62.7 ± 34.7	73.9 ± 70.9	0.59
Conversion	0 (0%)	2 (6.9%)	0.49
M-VARC procedural success ^a^	5 (20.0%)	3 (10.3%)	0.45
Modified procedural success ^b^	18 (72.0%)	20 (69.0%)	1
Mitral stenosis			<0.001
None	2 (8.0%)	21 (72.4%)	
Mild	15 (60.0%)	7 (24.1%)	
Moderate	8 (32.0%)	1 (3.4%)	
Severe	0 (0%)	0 (0%)	
Mitral regurgitation			<0.001
None	18 (72.0%)	5 (17.2%)	
Mild	7 (28.0%)	21 (72.4%)	
Moderate	0 (0%)	2 (6.9%)	
Severe	0 (0%)	1 (3.4%)	
Type of regurgitation			<0.001
None	21 (84.0%)	5 (17.2%)	
Transvalvular	1 (4.0%)	4 (13.8%)	
Paravalvular	3 (12.0%)	19 (65.5%)	
Both	0 (0%)	1 (3.4%)	
LVOT obstruction	1 (4.0%)	2 (6.9%)	1.00

^a^—Procedure success as defined by the Mitral Valve Academic Research Consortium criteria, ^b^—device success modified so that mean gradient ≤ 10 mmHg was considered a success instead of ≤5 mmHg.

**Table 4 jpm-13-00803-t004:** Postoperative course and outcomes.

	Valve-in-Valve (*n* = 25)	Valve-in-Ring (*n* = 29)	*p*-Value
Pacemaker implantation	0 (0%)	1 (3.5%)	1.00
Acute kidney injury	6 (24.0%)	6 (20.7%)	1.00
New-onset dialysis	1 (4.0%)	5 (17.2%)	0.20
Wound-healing disorder	0 (0%)	0 (0%)	1.00
Sepsis	2 (8.0%)	3 (10.3%)	1.00
Stroke	2 (8.0%)	1 (3.4%)	0.59
TIA	0 (0%)	1 (3.4%)	1.00
CPR	3 (12.0%)	2 (6.9%)	0.65
Ventilation time			1.00
Under 12 h	0 (0%)	0 (0%)	
Under 24 h	23 (92.0%)	25 (86.2%)	
Over 24 h	1 (4.0%)	1 (3.4%)	
RBC transfusion (units)	0.7 ± 2.0	2.3 ± 3.8	0.02
ICU stay (days)	3.8 ± 6.8	4.3 ± 6.3	0.96
Hospital stay (days)	9.9 ± 5.9	13.5 ± 8.0	0.13
30-day mortality	1 (4.0%)	2 (6.9%)	1.00
Overall mortality			
Cardiac death	5 (38.5%)	12 (52.2%)	0.66
Median survival time (years)	4.2	1.2	0.01

TIA—transitory ischemic attack, CPR—cardiopulmonary resuscitation, RBC—red blood cell, ICU—intensive care unit, LVOT—left ventricular outflow tract.

**Table 5 jpm-13-00803-t005:** Univariate risk factors for mortality.

	Hazard Ratio	95% CI	*p*
STS-PROM	1.07	0.99–1.15	0.08
EuroSCORE II	1.04	1.00–1.08	0.06
Age	1.02	0.97–1.08	0.38
Sex (female)	2.14	1.07–4.27	0.03
BMI	1.03	0.97–1.09	0.39
Diabetes on insulin	1.68	0.59–4.77	0.33
COPD	1.25	0.51–3.07	0.62
Pulmonary hypertension	2.13	0.75–6.06	0.16
Chronic kidney disease	1.65	0.83–3.26	0.15
GFR	0.99	0.97–1.00	0.14
Peripheral vascular disease	0.76	0.29–1.96	0.57
Atrial fibrillation	1.44	0.34–6.03	0.62
Preoperative pacemaker	1.77	0.92–3.43	0.09
Type of valve disease (functional)	1.55	0.80–3.00	0.19
Type of heart failure (HFrEF)	1.70	0.84–3.41	0.14
Baseline LVEF	0.98	0.96–1.00	0.10
History of stroke	1.12	0.46–2.7	0.80
Time since initial operation	1.07	0.99–1.16	0.11
Procedure duration	1.02	1.01–1.03	<0.001
Procedure type (ViR)	2.36	1.19–4.67	0.01
Valve size	0.93	0.77–1.10	0.41
Conversion to sternotomy	29.33	4.82–178.4	<0.001
Transfemoral access	2.06	0.95–4.46	0.06

**Table 6 jpm-13-00803-t006:** Risk factors for mortality in multivariate Cox regression.

	Hazard Ratio	95% CI	*p*
Sex (female)	2.49	1.19–5.20	0.02
Conversion to sternotomy	22.49	3.47–145.92	0.001
Procedure type (ViR)	2.06	1.01–4.21	0.05
EuroSCORE II	1.06	1.07–1.11	0.03

## Data Availability

The data presented in this study are available on request from the corresponding author. The data are not publicly available due to ethical restrictions.

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
