# Peer review of "Long-Term Outcomes after Transcatheter Mitral Valve-in-Valve or Valve-in-Ring Procedures"

_jpm, 2023, doi:10.3390/jpm13050803_

Round 1

Reviewer 1 Report

In the study titled “Long-term outcomes after transcatheter mitral valve-in-valve or valve-in-ring procedures”, Wilbring et al described long-term results of 54 consecutive patients undergoing transcatheter Valve-in-Valve or Valve-in-Ring procedures for failed mitral bioprostheses or recurrence of mitral regurgitation after surgical repair.

The work is well conducted and provides new insight into the field of valve-in-valve procedure, reporting on long-term results after transcatheter interventions.

Nevertheless, some aspects need to be further elucidated, in order to better understand these findings.  

1) Please, report all variables for which association with all-cause death was evaluated by univariate and multivariate Cox regression analysis (e.g. does vascular access used represent a predictor of worse outcomes?).

2) Learning curve for valve-in-valve procedures is not negligible. Procedures included in the study had been performed over a time-lapse of 10 years. Please, report the number of procedures performed each year and investigate if there is a trend in favor of better results over the years.

3) Please, correct the following sentences: lines 124 to 126 “The calculated risk scores were consistently higher in the ViR group with an EuroSCORE II averaging 11.5 ± 7.5% and a mean STS-PROM Score of 5.9 ± 3.7% compared to 16.9 ± 9.0%, and 8.7 ± 4.3% in ViV-patients (both p=0.01)”.

Values of EuroSCORE II and STS-PROM scores had been inverted between ViR and VIV patients.

Author Response

Dear Reviewer #1,

thank you for your constructive remarks. We hope to have them addressed adequately. Please see the attached file for a point-to-point response.

Sincerely

MW

Reviewer 2 Report

The authors of this manuscript sought to evaluate clinical outcomes among patients receiving percutaneous implantation in the mitral position of balloon-expandable transcatheter heart valves intended for the aortic valve within failed surgical rings or bioprosthesis. The manuscript is well-written and the subject is of great clinical interest. However, certain limitations should be acknowledged. My specific comments are as follows:

Introduction: Please avoid definite sentences “like the present study sought to close the long-term data gap by adding a nearly 10-years follow-up..”. The actual mean follow-up was way below 10 years, and the results of the study are not definite ought to sample size and statistical power. Authors should temper down this sentence.

METHODS: Main inclusion criteria should read as “recurrent MR after mitral repair using a surgical ring", since there are plenty of repairment techniques in the surgical field (e.g., neo-chords, etc) which do not involve the implantation of a mitral ring.

METHODS: The 1st paragraph of below the "patients, study groups and follow-up" heading (lines 90-94) seems more a description of the results than a summary of the methods used for the investigation.

Results: Why do the authors think there´s such a difference in prior permanent pacemaker rates between study groups? Do they think the higher rates of LV dysfunction and dilation for the ViR group may have played a role?

Figure 1: This figure is misleading. The study includes exclusively patients who underwent a re-intervention, hence it does not reflect the actual differences between surgical mitral valve replacement vs repair. We cannot deduce that both approaches have similar failure rates over time by the fact that there were no differences in this 54-patient population.

Table 2: there are 4 subcategories for RV function, and there are 5 rows of values.

Table 3: How it comes that moderate and severe mitral stenosis were each present in 32% of the patients (64% in total) in the ViV group, and the indication for redo procedure was stenosis just in 28% of the ViV patients?

Table 3: What access is right anterolateral? Please explain.

Discussion: lines 279-282. Such definite conclusions should be avoided. The present study lacks statistical power and baseline characteristics were importantly unbalanced between study groups. The present study is in line with prior results, but it does not confirm that ViR poses a poorer prognosis independently of confusion factors.

Discussion: The authors did not include a limitations section. This is important, considering the observational and retrospective nature of the investigation. The lack of statistical power, the low number of patients remaining when follow-up extends beyond 4 years, as well as the poor comparability of study groups due to baseline unbalanced features are the main aspects to highlight in this section.

Conclusions: temper down the conclusions as they stem from an observational study with a relatively poor sample size.

Author Response

Dear Reviewer #2,

thank you for your constructive remarks. We hope to have them addressed adequately. Please see the attached file for a point-to-point response.

Sincerely

MW
